# The Medical Relevance of *Fusarium* spp.

**DOI:** 10.3390/jof6030117

**Published:** 2020-07-24

**Authors:** Herbert Hof

**Affiliations:** MVZ Labor Limbach und Kollegen, Im Breitspiel 16, 69126 Heidelberg, Germany; herbert.hof@labor-limbach.de; Tel.: +06-221-34-32-342

**Keywords:** *Fusarium*, infections, antimycotics, mycotoxins, mycobiome, trichothecenes, mycoprotein

## Abstract

The most important medical relevance of *Fusarium* spp. is based on their phytopathogenic property, contributing to hunger and undernutrition in the world. A few *Fusarium* spp., such as *F. oxysporum* and *F. solani*, are opportunistic pathogens and can induce local infections, i.e., of nails, skin, eye, and nasal sinuses, as well as occasionally, severe, systemic infections, especially in immunocompromised patients. These clinical diseases are rather difficult to cure by antimycotics, whereby the azoles, such as voriconazole, and liposomal amphotericin B give relatively the best results. There are at least two sources of infection, namely the environment and the gut mycobiome of a patient. A marked impact on human health has the ability of some *Fusarium* spp. to produce several mycotoxins, for example, the highly active trichothecenes. These mycotoxins may act either as pathogenicity factors, which means that they damage the host and hamper its defense, or as virulence factors, enhancing the aggressiveness of the fungi. Acute intoxications are rare, but chronic exposition by food items is a definite health risk, although in an individual case, it remains difficult to describe the role of mycotoxins for inducing disease. Mycotoxins taken up either by food or produced in the gut may possibly induce an imbalance of the intestinal microbiome. A particular aspect is the utilization of *F. venetatum* to produce cholesterol-free, protein-rich food items.

## 1. Introduction

Several *Fusarium* spp. are important phytopathogenic fungi causing enormous economic loss in agriculture because of worldwide crop damages and wastages in livestock farming [1]. Hence, these fungi are contributing to the most serious medical problems of mankind, namely hunger and undernutrition. Indeed, certain Fusarium spp. can infect small grain cereals (wheat, barley, and oat) (for example, *Fusarium graminearun*) and maize (for example, *Fusarium verticillioides*), or many other plants. A particular label of *Fusarium* spp. is yet their ability to produce several mycotoxins in large amounts [1,2], endangering not only the health of animals but also that of humans. These fungal properties, however, are fairly neglected by medical mycologists. In principal, mycotoxins can act as pathogenicity factors, but sometimes also as virulence factors [3], which means that mycotoxins can either damage organs and in particular, the defense mechanisms of a host and/or may enhance the aggressiveness of a fungus.

Medical mycologists are primarily interested in the pathogenicity of fungi. In general, they look upon their ability to infect a human host. However, only a few out of the large list of *Fusarium* spp., in particular *Fusarium oxysporum* and *Fusarium solani* and more seldomly, *F. dimerum* and few others, are able to infect humans, at least in the case that the conditions in these hosts are favorable; this means that these opportunistic fungi are occasionally able to penetrate into the host, to spread and to induce an inflammatory reaction; in principle, these opportunistic pathogens are able to induce local and, rarely, systemic infections, especially in immunocompromised patients [4].

## 2. Biology

Fungi of the genus *Fusarium* belong to the group of ascomycetes. Since the hyphae of these molds are not pigmented, one could range them also among the hyalohyphomycetes. There are at least 70 defined and well described *Fusarium* spp., but possibly there are even much more species which are still not well characterized [2].

According to their teleomorphic descriptions, they belong mainly to two genera, namely to *Gibberella* and *Nectria* (Table 1). Among the various species, only a few of them are human pathogenic [4,5]; some examples representing the most relevant species are given in Table 1. Typically, they produce a lot of mycotoxins; the most relevant and the best described products are the trichothecenes, the zearalenones, and the fumonisins (Table 1).

*Fusarium* spp. are characterized by well developed, septated, non-pigmented hyphae with acute-angled bifurcations forming typical macroconidia, so-called sporodochial conidia [5]—varying in shape, size, and number from one species to another. One representative example is given in Figure 1, showing the macroconidia of *F. oxysporum*. The microconidia are so-called aleuroconidia, which do not originate from specialized conidiophores, but directly from the hyphae [6]. Sexual reproduction is rarely observed.

They grow rather rapidly on various common nutrient agars, for example, on potato dextrose agar or Sabouraud agar. The growth of *F. solani* is somewhat slower than that of *F. oxysporum*. The colonies are white and floccose. On the underside of the Petri dishes grown with *F. oxysporum*, one can sometimes see a red color in the agar layer shining through the colonies (Figure 2), whereas *F. solani* produces a blue-grey or brownish pigment.

An exact differentiation can be obtained either by MALDI-TOF MS [7] or even better, by molecular biologic methods such as PCR [8,9]. This is sometimes necessary, because the various species of *Fusarium* are located within complexes, composed of several species closely related to each other. For example, *F. solani* is composed of a complex consisting of >40 different species [5]. In the *Gibberella fujikuroi* complex, at least 27 species can be differentiated [10]. Recently, the whole genome of human pathogenic strains of *F. oxysporum* has been sequenced [11]. Obviously, in contrast to plant pathogenic strains, these fungi are specifically enriched in genes encoding virulence and resistance factors, which will enable these pathogens to adapt to special niches such as a human host.

### 2.1. The Role of Mycotoxins

*Fusarium* spp. naturally infect several plants. During infections of the ear and kernels of maize with *F. verticillioides* and of wheat, barley, and oat with *Fusarium graminearum*, large amounts of mycotoxins are produced before harvest. This occurs in practically all countries of the world. Furthermore, toxins are present in various fruits (for example, pineapples) and other vegetable feedstuffs. This fact is a source of grave concern for healthy nutrition, since some of these phytopathogenic fungi are able to produce a spectrum of mycotoxins. Fusarial mycotoxins are by definition secondary metabolites produced by toxigenic fungi. The most common groups are trichothecenes, zearalenones, and fumonisins (Table 2). From each of these mycotoxins, several derivatives are described, with more or less toxic potencies [1]. For example, the trichothecene group consists of at least 170 different compounds, such as, HT2, T2, diacetoxyscirpenol, deoxynivalenol (DON), nivalenol, fusarenon X, 3-ADON, and 15-ADON. In addition, certain *Fusarium* spp. produce several other so-called emerging mycotoxins, i.e., beauvericin, butenolides, culmorin, enniatins*,* fusaric acid, fusarin C, moniliformin, etc. The various biological effects of these mycotoxins (Table 2) have been described in detail elsewhere [12,13]. In general, the trichothecenes T2 and nivalenol are more toxic than DON towards animals and man, while DON is more toxic for plants. Nivalenol is a potent inhibitor of protein, RNA, and DNA synthesis in mammalian cells and can cause necrosis of cells, especially in tissues that are rapidly growing and dividing, such as intestinal epithelial cells [1].

In general, one fungal species can produce several mycotoxins. For example, F. graminearum is able to produce 3-ADON, 15-ADON, butenolide, culmorin, deoxynivalenol (DON), diacetoxyscirpenol, fusarenon-X, nivalenol, and zearalenones [12]. On the other hand, one has to keep in mind that individual strains may have lost the capacity for mycotoxin production. Furthermore, the amount of toxin produced definitely depends on the growth conditions, such as temperature, substrates, time of incubation etc. Hence, it is difficult to predict the toxic burden of a food item; therefore, if desired, it can be measured by proper laboratory methods, such as ELISA or HPLC.

Acute intoxications are rare in developed countries so that clinicians become unaware of the health risks of these mycotoxins. Historic reports [14] about mass casualties are known, for example, in Russia in the 1930s and 1940s, when Fusarium-contaminated bread containing T2 mycotoxin was distributed, causing alimentary toxic aleukia with a 60% mortality rate. Symptoms began with vomiting, abdominal pain, (bloody) diarrhea due to gastrointestinal ulcerations, and prostration. Some days after, the consumers complained of fever, chills, and myalgias. Further symptoms included pharyngeal or laryngeal ulceration and diffuse bleeding into the skin. Bone marrow depression occurred, leading to granulocytopenia and even pancytopenia. These immunocompromised patients suffered from secondary sepsis.

Much more relevant in practically all parts of the world, however, seems to be the chronic exposition of individuals to fusarial mycotoxins, although the exact role of one or a combination of fusarial mycotoxins is difficult to clarify because a quantitative designation of toxin intake along decades is retrospectively hardly to be quoted. Furthermore, the health detriments, such as cancer, are so non-specific and multifactorial that several other causes are possibly involved. Hence, the attributable dimension of mycotoxins in food to any health injuries remains largely undefined.

There are reports that some of the trichothecenes dispose of immunomodulatory activities (Table 2), leading to exacerbations and aggravations of viral as well as bacterial infections, such as salmonellosis [15]. Furthermore, *Fusarium* mycotoxins may affect the morphology and the barrier function of the intestinal layer, leading to increased translocation of different bacterial species to the systemic compartment [1,15,16,17].

It is worth noting that exposure of pregnant women to fumonisin has been accused to interfere with the neural tube development of the fetus, resulting in spina bifida and anencephaly through interference with the function of folate binding protein and the utilization of folic acid [16,18].

### 2.2. The Role of Infections

#### 2.2.1. Clinical Manifestations

Since *Fusarium* spp. are widespread in nature, and fungal conidia are distributed by air, saprophytic colonization of skin and respiratory mucosa is common. Even in the gut flora, they are found among many other bacteria and fungi [19]. Human infections with *Fusarium* spp., however, are rare [6,20]. The most frequent fungal species found are *F. oxysporum* and *F. solani*. Yet, many other *Fusarium* spp., especially *F. dimerum* and *F. guttiforme* (as well as other strains of the *Fusarium fujikuroi* complex), are found occasionally. Obviously, these opportunistic pathogens wait for their chances, since in most cases, a traumatic injury or an immunocompromised state precede infection.

Manifestations of *Fusarium* infections are quite different (Table 3).

##### —Local infections

*Fusarium* spp. are occasionally isolated from nails. Whereas there is no doubt that these fungi, although they do not degrade keratin, are, in principle, able to trigger onychomycoses, the pathologic role of these isolates has to be questioned, however, in an individual case, since a mere colonization or contamination, respectively, has to be considered also. Local skin infections can present as subcutaneous nodules following dissemination [21,22], but occur also in preexisting skin lesions, when the epithelial barrier has been damaged and open wounds are present. Extended burn wounds tend to be infected in time with molds including *Fusarium* spp.

After traumatic inoculation or iatrogenic injury of the eye, the fungi may induce keratitis or even endophthalmitis [8]. Since *Fusarium* spp. are able to produce biofilms on contact lenses [23], it is quite conceivable that keratitis is often associated with contact lens carriage [24]. Deep ulceration and even perforation of the cornea can be observed in those cases, leading to an inflammatory infiltration of the anterior chamber and sometimes, to overt endophthalmitis requiring irrigations, a keratoplasty, or finally, an enucleation of the infected eye.

Concomitant with other manifestations, invasions of the maxillary and ethmoidal sinuses were observed often [25]. Continuous ambulatory peritoneal dialysis (CAP) represents a risk of portal of entry of environmental fungi, causing fusarial peritonitis [26].

##### —Systemic fusariosis

These mold infections are rare and sporadic, but severe and often life-threatening. They will affect predominantly granulocytopenic patients. Those manifestations are associated with a very high mortality rate, ranging from 50% to 80% [20]. Since *Fusarium* spp. are able to form within tissue microconidia, which can be distributed hematogenously [6], dissemination into various organs can occur [27], in particular, the lungs. Pneumonia is a common manifestation during systemic fusariosis in neutropenic patients [6,25]. The clinical manifestations and consequences are quite similar to those of an aspergillosis.

During dissemination, *Fusarium* spp. can adhere to foreign bodies (indwelling catheters, artificial heart valves, joint replacements) and afterwards, grow in biofilms on the surface of these materials, which are difficult to cure by antimycotic therapy.

#### 2.2.2. Diagnosis

Local infections of the skin can be recognized easily by inspection. Eye infections are documented by ophthalmologic procedures, by microscopic examination of scrapings, and by culture. Infections of the maxillary and ethmoid sinuses can be suspected in the case of presence of hyperdense opacifications [25]. Fungal pneumonia is detected by imaging, such as CT chest scans, characterized by hypodense infiltrates without halo signs and without vessel occlusion signs. In many instances, these presentations are indistinguishable from aspergillosis [6,25].

Galactomannan assay, which is often positive in aspergillosis patients, can give positive results during fusariosis, too [6]. Thus, this laboratory method is not able to differentiate the two clinical entities and is, thus, a suitable, but not discriminative, laboratory test for the approval of fusarial infections.

Cultures, either from superficial lesions, from nasal sinuses or from BAL, represent the mainstay in diagnosis of *Fusarium* infections. In contrast to other mold infections, blood cultures can become positive during infections with Fusarium spp. [21], because these fungi form aleuroconidia during multiplication in host tissues [6]. The colony morphology (Figure 2), as well as the micromorphological characteristics of segmented, acute-branched hyphae, and typical macroconidia (Figure 1), can help at least highly experienced personnel to identify a definite species. The characteristic structures are fully developed after about seven days. For exact species diagnosis, in general, a molecular characterization is advisable, however [8,9]. Although, in principle, MALDI-TOF MS seems also to be a suitable and accurate technology for the identification and differentiation of the various *Fusarium species* [7], this technique is not yet used routinely.

In vitro testing of the susceptibility of *Fusarium* spp. to antimycotics is not a standard procedure, but MIC values can be obtained according to recommendations [9]. In general, amphotericin B and voriconazole display the lowest MIC values. Posaconazole and isavuconazole seem to be less active [9]. ECOFF values have to be used for interpretation instead of given, approved breakpoints.

The profile of mycotoxins produced by fungi isolated from food can be monitored in the laboratory. In medical microbiology, however, testing of mycotoxin production of isolated *Fusarium* spp. does not play any practical role, although the isolates may differ considerably, which may possibly alter their virulence and their pathogenic potency.

#### 2.2.3. Therapy

Large clinical studies on the treatment of fusarium infections are still lacking. Liposomal amphotericin B and voriconazole have been used successfully [20]. In vitro, these antimycotics show the lowest MIC values [9]. In the genome of human pathogenic *F. oxysporum*, one can find a doubling of gene coding for ergosterol synthesis and, in addition, more than 70 copies of genes coding for various efflux pumps [11], which can at least partially explain the relative resistance to azoles. Breakthrough fusarioses have been reported in patients while on posaconazole prophylaxis [20]. Clinical data on the efficacy of isavuconazole are still lacking. In severely ill patients, combination therapy with liposomal amphotericin B plus voriconazole may be worth trying. To further improve the therapeutic outcome, a combination with terbinafine has been proposed [21]. In leukemic patients, recovery from granulocytopenia is most critical for a response to antifungal therapy. For bridging of neutrophil recovery, granulocyte transfusion has been successfully used in persistently granulocytopenic patients. Surgical debridement of necrotic tissues and removal of colonized central venous lines might additionally help [20], and furthermore, eliminate a potential portal of entry.

#### 2.2.4. Prophylaxis/Prognosis

Disseminated fusariosis may recrude when an immunosuppressed patient had been effectively treated in the first instance, but immunosuppressive therapy continued. It has been observed in mice that Fusarium may persist in the form of thick-walled (intercalar) chlamydospores in tissues, because these structures may be more resistant to the host’s defense system than hyphae [11]. Hence, an antimycotic secondary prophylaxis, for example, with voriconazole, should be considered for those patients, because of an otherwise poor prognosis [28].

### 2.3. The Role of Mycoprotein Production

For the commercial production of vegetarian meat, a particular strain of *F. venetatum* is used [29,30]. The dry mass of *F. venetatum* strain A3/5 (ATCC PTA-2684) contains 25% cell wall, consisting mainly of mannans, glucans, and to a minor extent, chitin, 48% protein, 12% soluble carbohydrate, and 12% fat. No cholesterol is found. The total protein content of the final product varies from 43–85%. Whereas these *Fusarium* spp. are not qualified to produce the most dangerous fusarial mycotoxins, some others, however, can be generally produced such as culmorin [31], with phytotoxic properties, weak cytotoxicity, and weak teratogenicity [12] and fusarin C [32], endowed with estrogenic agonist activity and carcinogenicity [12]. In addition, this Fusarium sp. is able to produce trichothecenes [12], such as diacetoxyscirpenol (cytotoxic, endocrine disruption, immune modulation, developmental and reproductive toxicity, and genotoxicity) [31,32] and even four other trichothecenes [29]. This particular strain used for industrial fabrication produces less mycotoxin than other strains of *F. venetatum*. No mycotoxins were detected in the final product destinated for consumption [33].

Rarely, allergic reactions against *F. venetatum* products have been observed [34].

## 3. Discussion

Systemic infections of immunocompromised patients with *Fusarium* spp. are rare but serious. In several aspects, they pass like other mold infections, such as aspergilloses and mucormycoses, so that one has to be aware of *Fusarium* infections in immunocompromised patients as a differential diagnosis [6,20]. Pneumonia is a frequent manifestation of disseminated fusariosis (Table 3). Halo signs and vessel occlusion signs are lacking, however. In contrast to aspergillosis, a blood culture can become positive. The test for galactomannan in BAL or serum is rather often positive [6]. Local infections of nails, skin, maxillary and ethmoidal sinuses, and eyes are further manifestations (Table 3) [8,24,25,26].

Out of the large number of *Fusarium* spp., only a few, such as *F. oxysporum* and *F. solani*, are human pathogenic (Table 1). The pathogenesis of infections with these *Fusarium* spp. remains still fairly unknown. Obviously, the human pathogenic strains yield special genes not present in plant pathogenic strains, enhancing their adaptation to the situation in humans, so that the fungi become more virulent. Namely, the production of proteins binding metal ions, such as iron or copper, will protect the fungi against oxidative stress [11]. In most cases, local damage of the epithelial barrier will be a prerequisite for the penetration of these opportunistic fungi. Furthermore, an immunodeficiency, especially a leukopenia, will further favor dissemination into various organs.

Although these fungi are not particularly aggressive in humans, the prognosis of infections, especially of systemic manifestations, is poor. This is partially due to the fact that these fungi are a priori resistant to many antimycotics, in particular, to azoles. Human pathogenic strains yield, indeed, genes encoding for resistance because of overproduction of targets and of efflux pumps [11].

It is well known that *Fusarium* spp. producing a lot of different mycotoxins are using these properties as pathogenicity and virulence factors to infect and damage various plants [35]. The role of mycotoxins as virulence factors in human infections, however, remains largely obscure. Indeed, for example, *F. oxysporum* can produce several mycotoxins, such as beauvericin (cytotoxic; antimicrobial activity), diacetoxyscirpenol (cytotoxic, immune modulatory, genotoxic), enniatins (cytotoxic, antimicrobial activity), fusaric acid (neurotoxic, phytotoxic, antibacterial activity), fumonisin (see Table 2), moniliformin (cytotoxic), and zearalenones (see Table 2) [12,36]. *F. solani* is known to secrete at least fusaric acid and fumonisin [12]. At the moment, it can only be speculated whether these mycotoxins may also play a pivotal role in the pathogenesis of human fusarial infections. In principle, it has been shown that fusarial mycotoxins are able to damage epithelial barriers [15,17,19] and can produce necrosis in tissues [1]; it can be anticipated that these cytotoxic as well as the immunotoxic substances (Table 2) could help the fungus to damage host tissues and to favor spreading. Because mycotoxins in general and fusarial mycotoxins in particular show pronounced synergistic effects [37], it could be awaited that their role is higher than previously suspected. The production of proteases may further contribute to the invasion into tissues [8]. If really a mycotoxin producing fungus profits from this property, then, these mycotoxins of *Fusarium* spp. act as virulence factors, like for example, what gliotoxin does in infections with *Aspergillus* spp. [3]; at the moment, this can be assumed but is not yet proven.

There are at least two sources of infections with *Fusarium* spp., namely the environment and the host’s own gut, which is regularly colonized with *Fusarium* spp. among other ascomycetic and basidiomycetic fungi and bacteria [19]. Until now, the endogenous source is hardly respected in medicine. Whereas it is well accepted that a protective isolation of immunocompromised patients by solitary confinement can prevent, to a certain extent, the exposure of a patient to external *Fusarium* conidia and reduce the risk of infection, a complete elimination of the risk of infection is by this way, however, not possible, since, at least, the endogenous risk will remain and the gut flora are impossible to eradicate. This endogenous route of infection could explain why some patients acquire a fungal infection during the time of isolation.

A mutual interaction between mycobiome and microbiome in the gut can be assumed [19], but the relevance is still largely unknown. Until now, little is known about the metabolic activities of fungi of the intestinal mycobiome and it remains obscure in particular whether mycotoxins are produced at all by *Fusarium* spp. residing in the gut and if applicable whether this activity plays any role in the pathogenesis of fusarial infections. It can, however, be taken for certain that fusarial mycotoxins taken up by various food items may alter the microbiome of a host, partially due to their antimicrobial activities [12]. In addition, mycotoxins may play a significant role in the defensive strategies of mycotoxigenic fungi against the resident microbes by quite other, different mechanisms, too [12]. Finally, this may lead to an imbalance. The extent will, however, highly depend on the amount of mycotoxins taken up, which is difficult to assess in each individual situation. Furthermore, the barrier function of the gut wall may be hampered by fusarial mycotoxins taken up by contaminated food [1,15,16,17], favoring the translocation of bacteria.

The potential clinical role of fusarial mycotoxins in food items as pathogenicity factors is well known [14]. One has to admit that today, the risk of acute intoxications of humans is low, at least in developed countries, where the amount of mycotoxins in food items is controlled by authorities. In veterinary medicine, however, episodes of acute intoxications are reported rather often [13]. Much more important but even less documented are the consequences of chronic exposure; because of the strong synergistic effects of mycotoxins in general [37], the impact on health of *Fusarium* spp. producing several mycotoxins at the same time must be even more pronounced.

One particular aspect in the medical relevance of *Fusarium* spp. is the use of *F. venetatum* to produce food items designated for human use, such as Quorn^TM^. It looks like meat, it tastes like meat, but it does not contain animal lipids such as cholesterol. Obviously, the mycotoxin production of the special *Fusarium* strain used is rather low and plays no practical role [31,32,33]. Furthermore, allergic reactions are quite rare [34]. This mycoprotein supply will presumably not be able to solve the problem of hunger and undernutrition in the world, which is caused, at least in part, by phytopathogenic *Fusarium* strains and by the production of fusarial mycotoxins.

## Figures and Tables

**Figure 1 jof-06-00117-f001:**
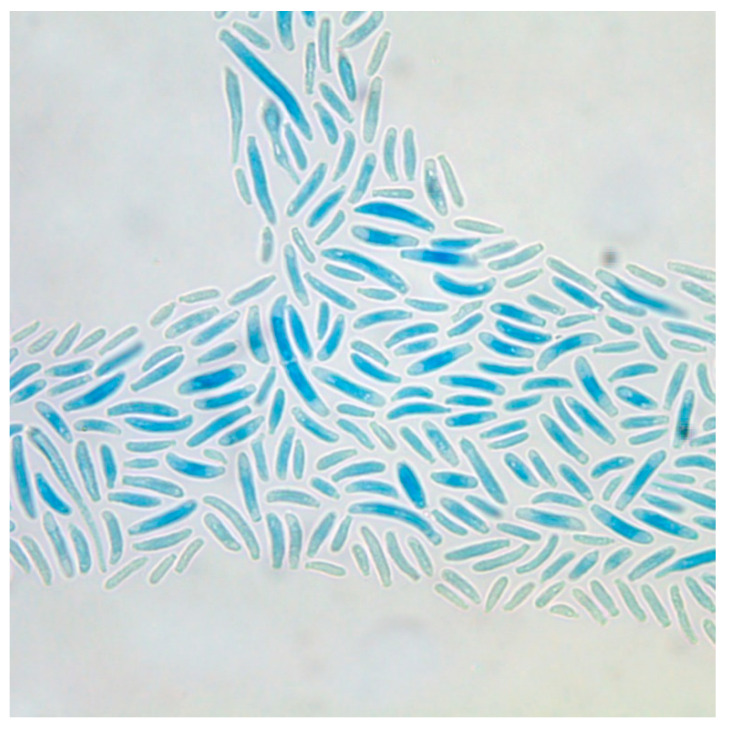
Micromorphological characteristics of *Fusarium* spp.: curved macroconidia of *Fusarium oxysporum.* (Magnification 400×).

**Figure 2 jof-06-00117-f002:**
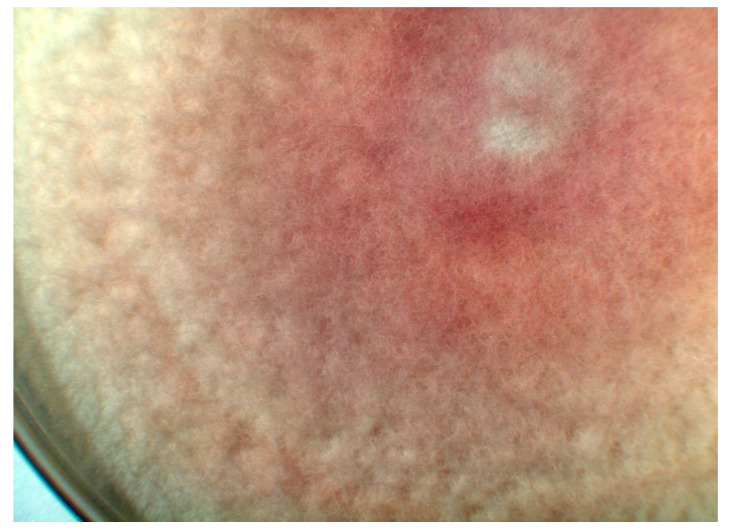
A floccose colony of *Fusarium oxysporum* on Sabouraud agar, seven days after incubation at 26 °C. The colonies themselves are white but they produce a red pigment, which is secreted into the surroundings. One sees the red color of the agar shining through. (Magnification 10×).

**Table 1 jof-06-00117-t001:** Characteristics of some relevant *Fusarium* species, including their ability to produce some relevant mycotoxins.

Anamorph	Teleomorph	Relevance	Trichothecenes	Zearalenones	Fumonisins
*Fusarium oxysporum*	unknown	human pathogenic (opportunistic); plant pathogenic (i.e., chickpea)	yes	no	yes
*Fusarium solani*	*Nectria haematococca*	human pathogenic (opportunistic); plant pathogenic	no	no	yes
*Fusarium venetatum*	unknown	plant pathogenic	Yes *	no	no
*Fusarium graminearum*	*Gibberella zeae*	plant pathogenic (especially cereals)	yes	yes	no
*Fusarium verticillioides*	*Gibberella moniliformis*	plant pathogenic (especially corn)	yes	yes	yes
*Fusarium guttiforme*	*Gibberella fujikuroi* complex	plant pathogenic (especially pine apple)	no	no	yes

* diacetylscirpenol.

**Table 2 jof-06-00117-t002:** Some characteristics of the most important fusarial mycotoxins according to [12].

Toxin	Risks
Trichothecenes such as T2, nivalenol, deoxynivalenol (DON), diacetoxyscirpenol et al.	cytotoxicity, endocrine disruption, immune modulation, developmental and reproductive toxicity, genotoxicity
Zearalenones	hormone-like activities (xenoestrogens), immunotoxicity, hepatotoxicity, carcinogenicity, nephropathy, hematotoxicity
Fumonisins	carcinogenicity, neurotoxicity (neural tube), hepatotoxicity

**Table 3 jof-06-00117-t003:** Major clinical manifestations of infections with *Fusarium* spp. according to [26].

Eye: — keratitis — contact lens infections — endophthalmitis
Skin: — subcutaneous nodules (after traumatic inoculation); possibly portal of entry. — wound infections; possibly portal of entry
Nails: — onychomycosis
Nasal sinuses: possibly portal of entry
Peritonitis (following CAP): possibly portal of entry
Systemic infections (blood cultures can be positive): — almost every organ can be affected, especially lungs — disseminated infections, leading to several septic metastases — foreign body infections — catheter infections

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
