# Peer review of "The Medical Relevance of *Fusarium* spp."

_jof, 2020, doi:10.3390/jof6030117_

Round 1
Reviewer 1 Report
The objective of this manuscript is to review the medical relevance of the fungus Fusarium as mentioned in the title. Interesting informations are given on the role of mycotoxins and their possible impact on health. However, this review does not reach what we are waiting for a real state-of-the-art with medical pertinence.
Here are some of my major comments :
The text of the manuscript doesn’t match enough with the title. Indeed, the clinical description of fusarial infections as well as diagnostic strategies are too poorly described for a medical journal. Please find some examples :
- Clinical description stars with onychomycosis that are still debated. Furthermore, diagnostic criteria for onychomycosis in order to avoid confusion with colonisation without infection are not mentioned.
- Severe systemic diseases that are very severe and lifethreatening are not well described in terms of underlying conditions. The pulmonary disease is mentioned but the infection is systemic and what is importznt is the diffusion in blood with secondary metastasis on skin and other organs. A recent clinical case published in the New England Journal of Medicine proposes a best description. Regarding diagnosis, blood culture are essential and need to be number 1 with some explanation : yeast-forms observed in the direct examination of the blood culture and a filamentous fungi in culture.
- Traumatic injury and burns are not enough detailed.
- Keratitis outbreaks in contact lens wearers were a major concern during the last years.
- What about Fusarium dimerum and Gibberella fujikori species complex?
- Etc…
Regarding treatment, does the author mean Line 170 secondary prophylaxis ? BBecause there is no recommanded primary prophylaxis for Fusarium infections
Poor illustration : clinical lesions ? figure 1 is of poor quality, no figure on microscopic examination of cultures ?
This is more a paper on mycotoxins rather a clinical paper, but then, some basic informations on virulence are lacking.
We Don't know what is the difference between this submitted manuscript and Ref 6. Hof, H. Schimmelpilze Teil 4: Fusarium. MTA Dialog. 2020; 21: in press
Author Response
The objective of this manuscript is to review the medical relevance of the fungus Fusarium as mentioned in the title. Interesting informations are given on the role of mycotoxins and their possible impact on health. However, this review does not reach what we are waiting for a real state-of-the-art with medical pertinence.
It was not the aim of this manuscript to give an extensive review, which will not be read by clinicians. But some new aspects have been included in the revised text.
Here are some of my major comments :
The text of the manuscript doesn’t match enough with the title. Indeed, the clinical description of fusarial infections as well as diagnostic strategies are too poorly described for a medical journal. Please find some examples :
- Clinical description stars with onychomycosis that are still debated. Furthermore, diagnostic criteria for onychomycosis in order to avoid confusion with colonisation without infection are not mentioned. Altered and extended
- Severe systemic diseases that are very severe and lifethreatening are not well described in terms of underlying conditions. The pulmonary disease is mentioned but the infection is systemic and what is importznt is the diffusion in blood with secondary metastasis on skin and other organs. A recent clinical case published in the New England Journal of Medicine proposes a best description. Regarding diagnosis, blood culture are essential and need to be number 1 with some explanation : yeast-forms observed in the direct examination of the blood culture and a filamentous fungi in culture. Altered and extended
- Traumatic injury and burns are not enough detailed. extended
- Keratitis outbreaks in contact lens wearers were a major concern during the last years. new literature (on biofilms on contact lens and clinical cases) is added
- What about Fusarium dimerum and Gibberella fujikori species complex? added
- Etc…
Regarding treatment, does the author mean Line 170 secondary prophylaxis ? BBecause there is no recommanded primary prophylaxis for Fusarium infections inserted, furthermore a hypothesis for the survival of Fusarium in tissues is given
Poor illustration : clinical lesions ? figure 1 is of poor quality, no figure on microscopic examination of cultures ?
I personally do not dispose of pictures from: clinical cases; a figure on microscopic features is added; the photo from culturs of F. oxysposrum is quite typical showing the floccose structure (aerial mycelia) and the red colour from the agar layer shining through.
This is more a paper on mycotoxins rather a clinical paper, but then, some basic informations on virulence are lacking.
Since clinician are not aware about the role of mycotoxins in general and fusarial mycotoxins in particular, this part is somewhat longer than in most papers on medical relevance of Fusarium.
We Don't know what is the difference between this submitted manuscript and Ref 6. Hof, H. Schimmelpilze Teil 4: Fusarium. MTA Dialog. 2020; 21: in press the citation of this paper written in german has been omitted
Reviewer 2 Report
There are a few misspellings and errors:
Line 6: ...relevance ... are is based on
L13-14 The sentence should be rephrased
L26: ... worldwide (1).
L29 verticillioides
L40 ... Fusarium oxysporum nand Fusarium solani
L73 Fusarium spp.
L84-86 Why italized?
L87: ...animals and man...
L91-99: Why italized?
L127: manifestation
L142 ...Fusaroium...
L168: ...patients...
L244: it tastes...
Author Response
Comments and Suggestions for Authors
There are a few misspellings and errors:
Line 6: ...relevance ... are is based on
L13-14 The sentence should be rephrased
L26: ... worldwide (1).
L29 verticillioides
L40 ... Fusarium oxysporum nand Fusarium solani
L73 Fusarium spp.
L84-86 Why italized?
L87: ...animals and man...
L91-99: Why italized?
L127: manifestation
L142 ...Fusaroium...
L168: ...patients...
L244: it tastes...
All recommendations are accepted and the alterations inserted in the new text
Reviewer 3 Report
The article represents a good overview of the medical relevance of Fusarium species.
Minor points:
- the author compares in the text (line 59) the appearance of F. oxysporum and F. solani with reference of figure 1, but show only F. oxysporum in the figure. Why is description limited to those two species?
- the author suggests PCR to differentiate the Fusarium species (line 67). What about MALDI-TOF? Please discuss!
- the intrinsic drug resistance of Fusarium should be discussed more precisely
- the chapter of mycotoxins should refer to table 1, where the production of mycotoxins by different Fusarium species is listed. Why does table 1 concentrates on production of trichothecens, zearalenones and fumonisins? Please explain in the text.
- formatting of the table makes it difficult to understand. And the author should mentioned that the rows with the mycotoxins should indicate whether the species produce the mycotoxins.
- The dry mass of F. venetatum strain A3/5 (ATCC PTA-2684) contains 25% cell wall, 48% protein, 12% soluble carbohydrate and 12% fat. What component is “cell wall”?
- all species names should be written italic
- instead some mycotoxins and sentences are written italic (line 84ff). Why?
- line 150: “amphotericin B” instead of “amphotericin B”
- heading of table 1 and 2: “characteristics” instead of “characters”
Author Response
The article represents a good overview of the medical relevance of Fusarium species.
Minor points:
- the author compares in the text (line 59) the appearance of F. oxysporum and F. solani with reference of figure 1, but show only F. oxysporum in the figure. Why is description limited to those two species?
It has been explained
- the author suggests PCR to differentiate the Fusarium species (line 67). What about MALDI-TOF? Please discuss!
The literature is added
- the intrinsic drug resistance of Fusarium should be discussed more precisely
New informations are added
- the chapter of mycotoxins should refer to table 1, where the production of mycotoxins by different Fusarium species is listed. Why does table 1 concentrates on production of trichothecens, zearalenones and fumonisins? Please explain in the text.
These informations are given in the new text
- formatting of the table makes it difficult to understand. And the author should mentioned that the rows with the mycotoxins should indicate whether the species produce the mycotoxins. yes
- The dry mass of F. venetatum strain A3/5 (ATCC PTA-2684) contains 25% cell wall, 48% protein, 12% soluble carbohydrate and 12% fat. What component is “cell wall”?
explained
- all species names should be written italic ok
- instead some mycotoxins and sentences are written italic (line 84ff). Why?
This has been a mistake of the publisher. This was not my intentation
- line 150: “amphotericin B” instead of “amphotericin B” ok
- heading of table 1 and 2: “characteristics” instead of “characters” ok